# Nanoemulsion Co-Loaded with XIAP siRNA and Gambogic Acid for Inhalation Therapy of Lung Cancer

**DOI:** 10.3390/ijms232214294

**Published:** 2022-11-18

**Authors:** Minhao Xu, Lanfang Zhang, Yue Guo, Lu Bai, Yi Luo, Ben Wang, Meiyan Kuang, Xingyou Liu, Meng Sun, Chenhui Wang, Jing Xie

**Affiliations:** 1School of Pharmacy and Bioengineering, Chongqing University of Technology, Chongqing 400054, China; 2Chongqing Key Laboratory of Natural Product Synthesis and Drug Research, Innovative Drug Research Center, School of Pharmaceutical Sciences, Chongqing University, 55 South Daxuecheng Road, Chongqing 401331, China

**Keywords:** pulmonary delivery, co-delivery, RNAi, nanoemulsions, lung cancer

## Abstract

Lung cancer is a leading cause of cancer mortality worldwide, with a 5-year survival rate of less than 20%. Gambogic acid (GA) is a naturally occurring and potent anticancer agent that destroys tumor cells through multiple mechanisms. According to the literature, one of the most potent inhibitors of caspases and apoptosis currently known is the X-linked Inhibitor of Apoptosis Protein (XIAP). It is highly expressed in various malignancies but has little or no expression in normal cells, making it an attractive target for cancer treatment. Here we report the development of a chitosan (CS)-based cationic nanoemulsion-based pulmonary delivery (p.d.) system for the co-delivery of antineoplastic drugs (GA) and anti-XIAP small interfering RNA (siRNA). The results showed that the chitosan-modified cationic nanoemulsions could effectively encapsulate gambogic acid as well as protect siRNA against degradation. The apoptosis analysis confirmed that the cationic nanoemulsions could induce more apoptosis in the A549 cell line. In addition, most drugs and siRNAs have a long residence time in the lungs through pulmonary delivery and show greater therapeutic effects compared to systemic administration. In summary, this work demonstrates the applicability of cationic nanoemulsions for combined cancer therapy and as a promising approach for the treatment of lung cancer.

## 1. Introduction

As one of the most common malignant tumors in the world, lung cancer poses an enormous threat to human life and health [1]. Patients with early-stage lung cancer generally have no obvious symptoms, and most patients are already in the middle and late stages when detected, leading to poor treatment effects [2]. Currently, the main treatments for lung cancer are chemotherapy, radiotherapy, and surgery [3,4]. Although there have been some advances in diagnosis and treatment, the current shortage of effective first-line antitumor drugs, multidrug resistance of tumor cells [5], and unsatisfactory drug delivery methods [6,7] make improving the prognosis and survival rate of lung cancer patients much more difficult [7]. Consequently, the development of effective methods of treating lung cancer is imperative.

Traditional intravenous administration is a major obstacle to lung cancer treatment due to its inefficiency and side effects [8]. Compared to other administration routes, pulmonary delivery of drugs to treat lung diseases has clear advantages [7]. The accumulation of the drug in the lungs can be accelerated by topical administration [9,10]. Inhalation administration restricts the drug’s ability to spread throughout the body, thus lowering systemic toxicity [6]. In most circumstances, inhalation medication delivery is superior to oral or intravenous administration in terms of area under the curve (AUC), half-life, and drug concentration in the lungs [11].

Natural plant components are a significant source of chemotherapy medications and are thought to be less hazardous than the pharmaceuticals currently in use [12]. A naturally occurring anti-cancer substance known as gambogic acid (GA) is obtained from the Garcinia cambogia tree. In some malignancies, GA has been demonstrated to exhibit anti-angiogenic capabilities, antiproliferative effects, induce apoptosis, and reverse multidrug resistance (e.g., in breast and prostate cancer) [12,13]. Even though numerous in vitro and in vivo tests have indicated that gambogic acid has anticancer potential [14], its poor biodistribution, multiple targets, and low water solubility have restricted its practical applicability [13]. In earlier research, a variety of medication administration methods were used to increase GA utilization and lessen side effects. For the treatment of triple-negative breast cancer, Sang et al. described a redox/pH-responsive multifunctional composite magnetic micelle modified with hyaluronic acid that improved uptake into TNBC tumor cells, rapid drug release, and antitumor-enhanced tumor activity in vivo [15]. Liang et al. used hyaluronic acid to graft gambogic acid (HA-GA) and encapsulate the hydrophobic photosensitizer chloroprene Ce6 (Ce6) to prepare ROS-sensitive nanomicelles (HA-GA@Ce6). The results showed that the nanomicelles exhibit enhanced cellular uptake into 4T1 tumor cells and may produce greater anticancer efficacy by lowering intracellular antioxidant levels during tumor-specific triggered photodynamic therapy [16].

One of the most effective inhibitors of caspase and apoptosis found to date is the X-linked Inhibitor of Apoptosis Protein (XIAP) [17], which is also known to be critical for cell invasion, migration, and apoptosis [18,19]. It is an appealing target for cancer therapy since it is highly expressed in a variety of malignancies but is either absent or very poorly expressed in normal cells [17,20]. Many studies have demonstrated that antisense oligonucleotides or small interfering RNA (siRNA) can suppress tumor cell proliferation, cause apoptosis, and make tumor cells more sensitive to chemotherapeutic treatments by inhibiting XIAP expression [21,22]. At the moment, simultaneous delivery of nucleic acids and cancer-fighting medications via a single nanoparticle platform offers good therapeutic benefits [23,24]. In this scenario, systemic toxicity and the dose of cancer treatment would be reduced. Additionally, combination therapies created through the concurrent administration of anticancer medications and nucleic acids may have therapeutic benefits that are additive or synergistic [25]. As a result, the combination of GA and XIAP inhibition is a promising lung therapy for lung cancer.

In this study, we designed cationic nanoemulsions for the simultaneous delivery of siRNA and gambogic acid. Nanoemulsions increased the solubility of gambogic acid, and chitosan, a naturally occurring cationic polymer, was employed to coat the nanoemulsions’ surface. Chitosan can also interact electrostatically with siRNA to produce electrostatic complexes that increase the stability of siRNA in vivo and the effectiveness of cellular uptake (Figure 1A). Hence, using pulmonary administration and systemic injection, respectively, to transport the drug-loaded system into the lungs of lung cancer model mice, the biodistribution of the cationic nanoemulsions and the effectiveness of lung tumor inhibition were assessed (Figure 1B).

## 2. Results

### 2.1. Preparation and Characterization of Nanoemulsions

By using dynamic light scattering (DLS), cationic nanoemulsions’ mean particle sizes were determined to be 183.81 ± 1.15 nm, with a PDI of 0.210 ± 0.0.13 (Table 1). The size and PDI of the cationic nanoemulsions were altered to 174.53 ± 0.78 nm and 0.262 ± 0.014, respectively, after loading GA. In the transmission electron microscope (TEM), the emulsions appeared as regular spherical structures (Figure 1A) with a zeta potential of 9.93 ± 0.58 mV for the GA-CNE (Table 1). Since the nanoemulsions had a good solubilizing effect on insoluble drugs, the prepared nanoemulsions had a drug encapsulation rate as high as 91.51 ± 3.732% (Table 1). To study drug uptake in cells, C-6 (coumarin-6) was used as a fluorescent probe, and its particle size, potential and PDI were characterized (Table 1). To create GA-SNE, siRNA and GA-CNE were combined in varying quantities, and a laser particle size analyzer was used to determine the average particle size and zeta potential of the mixture. Figure 1C, D displays the results. The figure shows that when GA-CNE and siRNA were combined at a volume ratio of 1:1, the positive charge of chitosan was neutralized by the negative charge of siRNA, resulting in a zeta potential of −11.37 ± 0.62 mV and an increase in particle size. This might be the result of the system becoming unstable, since there was not enough chitosan in GA-CNE to completely load all of the siRNAs. The charge of GA-CNE returned to positivity as the volume rose, and the particle size likewise tended to stabilize. The same procedures were used to prepare GA-SNE with various weight ratios, and agarose gel retardation electrophoresis was used to determine the degree of binding between GA-SNE and siRNA. Figure 1B presents the findings. The graphic shows that free siRNA migrates during electrophoresis, and the band gradually darkens as the *w*/*w* value rises. There is no band in the lane when the GA-CNE: siRNA *w*/*w* value is 4. The bands and brightness are entirely focused in the dotted wells, showing that GA-CNE can completely bind to siRNA when GA-CNE: siRNA *w*/*w* = 4. C6-CNE was subsequently characterized according to the same conditions and showed similar results (Figure 1B). Additionally, we looked into the carrier’s stability. As seen in Figure 1E, GA-CNE maintained a constant size and PDI, which supported the carrier’s strong stability.

### 2.2. Cellular Uptake

Coumarin-6 was employed as a fluorescent probe to examine how GA-SNE is absorbed by cells, and A549 cells were given GA-SNE, free siRNA, or GA-CNE treatments, as appropriate. Flow cytometric analysis and confocal microscopy were utilized to measure the levels of various formulations integrated into the A549 cells after 4 h of incubation. Due to the positive charge and tighter size distribution, GA-CNE demonstrated a significant level of cellular absorption. The fact that GA-SNE was substantially more readily absorbed as compared to free siRNA absorption suggests that cationic nanoemulsions can bind to and safeguard siRNA. Further confirmation of such results was obtained using flow cytometry (Figure 2B,C). Once the siRNA enters the cell, it must exit the lysosome to post-transcriptionally silence cytoplasmic genes. By labeling the cell nucleus with 4-diamidino-2-phenylindole and the lysosomes with LysoTracker Green, the co-staining experiment was further carried out to investigate the endosomal escape of siRNA. As can be seen, free siRNA (red) colocalized with lysosomes (green), while part of the red fluorescence from GA-SNE started to dissociate from the green fluorescence (Figure 2D), implying that the siRNA started to migrate to escape from the lysosome. Results showed that GA-SNE successfully escaped endosomes and effectively delivered siRNA.

### 2.3. Cell Viability

In A549 cells, the ratio of GA to XIAP siRNA in GA-SNE was optimized. A549 proliferation was significantly inhibited by GA-SNE with a GA/siRNA ratio of 1:3 (Figure 3A). This ratio, therefore, was chosen for subsequent tests. A549 cells were treated with CS-CNE, CS-SNE, GA-CNE, and GA-SNE, respectively, to test the preparations’ in vitro anticancer effects. Cell viability was assessed using the CCK-8 cytotoxicity assay. Neither GA-CNE nor siRNA had a significant inhibitory effect on the cells. GA-SNE, on the other hand, showed the highest inhibitory effect among all groups (Figure 3B), indicating that the co-delivery system delivering siRNA and GA simultaneously would have a synergistic inhibitory effect on cells in vitro.

### 2.4. Analysis of Apoptosis and Cell Cycle Arrest

Using Annexin V-FITC labeling, apoptosis caused by GA and XIAP siRNA was identified. The untreated control cells showed a comparatively low percentage of apoptosis (0.71%), as illustrated in Figure 3C. Furthermore, consistent with CCK8 results, cells incubated in the GA-SNE group underwent up to 94.53% apoptosis compared to CS-SNE, 78.12% higher than the effect of CS-SNE. These results suggest that GA and XIAP siRNA can promote A549 cell death by inducing apoptosis. Following treatment with various preparations, flow cytometry was used to identify the A549 cell cycle. Our results (Figure 3D,E) showed that GA and XIAP alone led to cell cycle G0/G1 arrest, similar to those previously reported [26,27]. Encouragingly, A549 cells were more frequently arrested in the G0/G1 phase in the combination group. These results suggested that by altering the cell cycle, our formulation can prevent tumor cells from proliferating.

### 2.5. Western Blot

A549 cells were treated with naked siRNA, CS-SNE, and GA-SNE, respectively, to further assess the gene interference effectiveness of siRNA-loaded nanoemulsions in vitro. Results from Western blotting revealed that both CS-SNE and GA-SNE dramatically decreased the expression of XIAP protein in A549 cells when compared to control and naked siRNA (Figure 4A). This indicates that chitosan as a cationic shell protects siRNA well and increases its stability. It has been demonstrated that GA-SNE has an in vitro gene interference effect. ImageJ was used to quantify the data, and the results demonstrated that siRNA could be successfully encapsulated in the cationic nanoemulsions for in vitro gene silencing (Figure 4B).

### 2.6. Biodistribution

Using tracheal instillation or intravenous injection, the in vivo imaging system tracked the distribution of Cy3-siRNA-loaded nanoemulsions in various organs at various periods. One hour after administration, the majority of the drug’s concentration in the p.d. group was in the lungs, whereas the i.v. injection group’s GA-SNE showed less lung accumulation and more accumulation at the renal site. At 12 and 24 h, the i.v. group showed negligible lung distribution, and the majority of the fluorescence was found in the liver and kidney (Figure 5B), indicating that most of the siRNA in the iv group was blocked by the kidneys and cleared quickly. In contrast, the p.d. group retained a significant amount of siRNA in the lungs at 12 and 24 h.

### 2.7. The Codelivery System’s In Vivo Therapeutic Effectiveness

The antitumor efficacy of the drug in combination with XIAP siRNA was investigated in Balb/C mice with lung cancer. Every three days, the body weight of each mouse was recorded during the animal experiment (Figure 5D), and there was no significant change in body weight in any group, implying the safety of the delivery system and treatment method. Pictures of the lungs taken from Balb/C mice in various treatment and control groups are shown in Figure 6C. In the lungs of B16F10 melanoma-bearing mice treated with the PBS group, there were numerous melanoma lymph nodes, and the tumor practically filled the entire lung. Significantly, compared to the PBS group or the single delivery method, the lungs of mice treated with the co-delivery system displayed fewer tumor nodes. Additionally, the GA-SNE p.d. group had fewer and smaller lung nodules than the GA-SNE intravenous therapy group, demonstrating the superiority of p.d.

### 2.8. Histopathological Examination

Hematoxylin and eosin (H&E) staining was used to further confirm anticancer efficacy. Comparing the groups that received PBS or the single delivery system, significant tumor nodules were seen in the lungs; however, in the co-delivery system, tumor mass and number were significantly reduced, and numerous normal pores were visible. The major organs of mice treated with GA-CNE, CS-SNE, and GA-SNE also showed little obvious pathology when stained with H&E compared to animals treated with PBS (Figure 5E), which demonstrated the high biocompatibility and biosafety of the nanoemulsions.

### 2.9. TUNEL Assay and Immunohistochemistry

Using the TUNEL assay, the induction of apoptosis in model mice was further studied. Compared to the control group and the single drug groups, the co-administration system dramatically increased the number of apoptotic cells in the tumor tissue (Figure 6A). According to IHC analysis, the primary biochemical marker of tumor growth, Ki67, was found to be highly expressed in both the control and single drug groups. However, GA and siRNA treatment significantly reduced the expression of Ki67 in tumor tissue, showing greater anticancer effects than other groups (Figure 6B). The data above demonstrate that the combined administration group could achieve significant antitumor efficacy by inducing cell apoptosis and inhibiting cell proliferation in comparison to the PBS or single administration groups.

**Figure 6 ijms-23-14294-f006:**
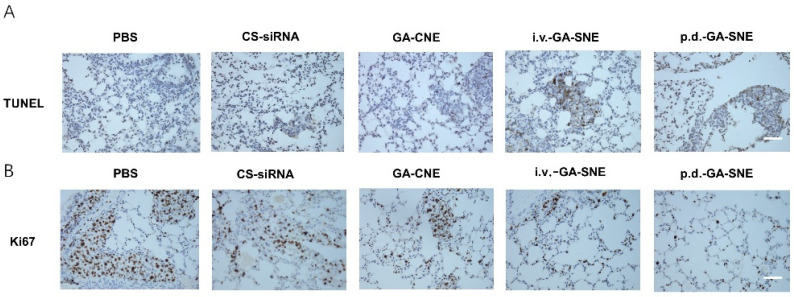
Analysis of murine tumor model treated with GA-CNE. (**A**) TUNEL analysis in the tumor region slices after all treatments. Scale bar: 100 μm (**B**) IHC analysis of Ki67 in the tumor region slices after all treatments. Scale bar: 100 μm.

## 3. Discussion

At present, nanoparticles show promise for inhalation drug delivery from the lung in the treatment of lung diseases. We successfully constructed a cationic nanoemulsion platform to achieve combined GA and siRNA therapy. However, pulmonary inhalation of nanoparticles is prone to receive clearance by mucous cilia in the lung, which affects the efficiency of drug pulmonary delivery, so it is necessary to further consider the movement and deposition of nanoparticles in the lung. In addition, the large surface area and rich vascularity of the lung lead to differences in the pharmacokinetics of the drug compared to systemic administration, so further studies are needed to investigate differences in drug metabolism due to differences in the two modes of administration. In addition, the toxicity of excipients is also a factor that should be considered for the safety of nanoformulations for inhalation administration. The excipients and carrier materials involved in the preparation process may lead to unintended adverse reactions, and therefore, for the application of nanoformulations to the lung, the potential pulmonary toxicity issues that may exist must be considered.

Therefore, to better enable pulmonary delivery of GA and siRNA combination therapy in clinical practice, more thorough studies are needed to optimize the composition, particle size, preparation methods, drug metabolism kinetics, inhalation loading compatibility and excipient safety of cationic nanoemulsions. These efforts will help to overcome the challenges of pulmonary delivery.

## 4. Materials and Methods

### 4.1. Materials

Gambogic acid was purchased from Sigma-Aldrich (St. Louis, MO, USA). Cy3-labeled siRNA (XIAP siRNA) was purchased from RiboBio (Guangzhou, China) with the following sequences: XIAP siRNA (sense): 5-AAG UGG UAG UCC UGU UUC AGC-3. Monoclonal antibodies (XIAP) and HRP-linked antibodies were purchased from Cell Signaling Technology (Danvers, MA, USA). Cell Count Kit-8, 4′,6-diamidino-2-phenylindole (DPAI), and LysoTracker Green were from the Beyotime Institute of Biotechnology (Shanghai, China). Annexin V-FITC Apoptosis Detection Kit and Cell Cycle and Apoptosis Detection Kit were from Beijing Biotopped Technology Co., Ltd. (Beijing, China). Tween-80, squalene, and glycerol were manufactured by Shanghai Energy Chemical Co., Ltd. Immunohistochemistry Kit (D601037-0050) was purchased from Sangon Biotech (Shanghai, China). Deionized water was produced in the laboratory. The reagent-grade chemicals used in the tests were all used exactly as provided to the researchers.

### 4.2. Cell Culture and Animal Care

A549 cells were bought from ATCC and grown in DMEM with 10% FBS supplement. All cells were kept in an incubator at 37 °C with 5% CO_2_. Fetal bovine serum (FBS), Trypsin and DMEM were bought from Hyclone (Logan, UT, USA). BALB/c female mice (18–20 g), 6–8 weeks old, were purchased from Byrness Weil Biotech Ltd. (Chongqing, China). The Chongqing University of Technology Experimental Animal Ethics Committee developed standards for laboratory animal care, which were followed in all animal research.

### 4.3. Preparation and Characterization of Nanoemulsions

By using a high-energy emulsification technique, nanoemulsions were created. To prepare the water phase, Tween 80 and glycerol (4:1, volume ratio) were added to PBS, while the oil phase was created by dissolving gambogic acid in a small amount of ethanol and combining it with squalene. Then, a coarse emulsion was prepared by gradually adding the oil phase while stirring continuously. Then, to prepare nanoemulsions (GA-NE), we homogenized the prepared coarse emulsion using a high-pressure homogenizer (Homogenizing Systems Ltd., Harlow, UK). We set the pressure setting to 100 Mpa and homogenized for a total of 10 cycles. The aqueous phase of the cationic nanoemulsions (GA-CNE) was supplemented with an additional 1% hydrochloric acid chitosan (70–90% deacetylation degree), and the remaining processes remained the same. C6-CNE was prepared by replacing GA with C6, and the rest of the process remained the same. The NanoBrook 90Plus PALS Zetasizer (Brookhaven, NY, USA) and transmission electron microscopy (TEM) (Hitachi, Japan) were used to determine the surface charge, size, and morphology of GA-CNE, respectively. Every day for 7 days at 37 °C, the changes in particle size and the polydispersity index (PDI) of GA-CNE in PBS were monitored to evaluate the stability. The amount of encapsulated GA in GA-CNE was measured at 360 nm using HPLC. The following equations were used to calculate the encapsulation efficiency:(1)$$\mathrm{Encapsulation}\;\mathrm{rate}(\%)\;=\frac{\mathrm{The}\;\mathrm{content}\;\mathrm{of}\;\mathrm{gambogic}\;\mathrm{acid}\;\mathrm{in}\;\mathrm{nanoemulsions}}{\mathrm{Dosage}\;\mathrm{of}\;\mathrm{gambogic}\;\mathrm{acid}}\;\times \;100\%\;$$

### 4.4. Preparation and Characterization of GA-SNE

To load XIAP-siRNA onto GA-CNE (GA-SNE), siRNA solutions (20 μM) were put into GA-CNE solutions at 1:1, 2:1, 4:1, 8:1, and 16:1 with *w*/*w* ratios of GA-CNE/siRNA and left at room temperature for 30 min. The development of the GA-CNE/siRNA complex was then verified using a gel retardation experiment employing 2% agarose gel and 100 V voltage for 15 min. The complexes’ size and zeta potential were determined using a NanoBrook 90Plus PALS Zetasizer (Brookhaven, NY, USA).

### 4.5. Cellular Uptake

Coumarin-6 was employed as the fluorescent probe and 5 × 10^4^ A549 cells were used to examine the uptake effect of nanoemulsions. In a 24-well plate, A549 cells were cultivated for the night. For 4 h, cells were exposed to 50 nM free Cy3-siRNA, GA-CNE (containing 0.05 μg/mL Coumarin-6), and GA-SNE (containing both 50 nM siRNA and 0.05 μg/mL Coumarin-6). After three PBS washes, the samples were fixed with paraformaldehyde (4%) for 10 min, stained with DAPI (2 μg/mL) for 10 min to highlight the cell nuclei, and then rewashed three times with PBS. A confocal laser scanning microscope (CLSM) was used to measure how well the nanoemulsions were internalized by the cells (Nikon, Japan).

#### 4.5.1. Quantitative Cell Uptake

The internalization of different agents in A549 cells was quantitatively evaluated by flow cytometry. Briefly, 24-well plates with 5 × 10^4^ A549 cells were planted and cultured at 37 °C for 24 h. After that, the culture medium was replaced with fresh medium containing Cy3-labeled siRNA, GA-CNE, and GA-SNE. After 4 h, the medium was withdrawn, and the cells were trypsinized, centrifuged, twice-washed in PBS, and then resuspended in 500 mL PBS for CytoFLEX FCM analysis (Beckman Coulter Corp., Brea, CA, USA). The control group consisted of cells that received no treatment.

#### 4.5.2. Endosomal Scape

The method used to prepare the samples for this study was the same as that described in Section 4.5 for cellular uptake. LysoTracker Green was used to label A549 cells after they had been exposed to the material for 4 h at 37 °C. For this, cells were given 50 nM of the dye diluted in DMEM, and it was left in place for 30 min. Following the removal of the dye, the cells underwent three PBS washes. The cells were then labeled with DAPI, fixed with 4% paraformaldehyde, and examined under confocal microscopy.

### 4.6. Cell Viability

A549 cells (1 × 10^4^ cells/well in a 96-well plate) were treated with GA-SNE with various molar ratios of GA: XIAP siRNA (3:1, 2:1, 1:1, 1:2, and 1:3) to optimize the ratio of GA and XIAP siRNA in GA-SNE. About 1 × 10^4^ A549 cells were grown in fresh media with various formulations and incubated at 37 °C with 5% CO_2_ to determine the toxicity of the samples. After 24 h, the samples were incubated for 4 h with a 10% CCK8 reagent in place of the culture medium. Utilizing a microplate reader to measure absorbance at 450 nm, the percentage of relative cell viability was calculated in comparison to that of the negative control (cells treated with PBS). For this test, four groups were used: GA-CNE (GA = 0.1−0.6μg/mL), GA-SNE (GA = 0.1−0.6μg/mL, ratios of GA: XIAP siRNA = 3), CS-SNE (with an equivalent amount of siRNA as GA-SNE) and CS-CNE (with an equivalent amount of chitosan as GA-CNE).

### 4.7. Evaluation of Cell Cycle Arrest and Apoptosis

Using the Cell Cycle and Apoptosis Analysis Kit (Top0232-50T, Biotopped, Beijing, China) and the ANNEXIN V-FITC/PI Apoptosis Assay Kit (TOP2210-100, Biotopped, Beijing, China), respectively, flow cytometry analysis was used to examine the induction of cell cycle arrest and apoptosis. About 5 × 10^4^ A549 cells were grown in a 24-well plate and incubated for 24 h at 37 °C. Fresh culture medium containing free siRNA, CS-SNE, GA-CNE, and GA-SNE was then used to treat the cells. The GA and XIAP siRNA doses per well were 0.3 μg/mL and 50 nM, respectively. After being incubated for 24 h, the cells were rinsed with PBS, collected, and resuspended in 500 L of binding buffer, then stained following the instructions provided with the apoptosis detection kit. Then, using flow cytometry, the fluorescence intensity of the labeled cells was evaluated. Cells were collected and preserved with 70% ethanol at −20 °C for the duration of the cell cycle test. Cells were stained for 5 min with PI and examined with a CytoFLEX FCM (Beckman Coulter Corp., Brea, CA, USA).

### 4.8. Western Blot

After being plated into a 6-well plate, about 5 × 10^6^ A549 cells were left to incubate overnight. Following a PBS wash, the transfection method using various formulations was carried out as previously mentioned. Total protein was then extracted after 24 h, separated using SDS-PAGE, transferred to a nitrocellulose membrane, probed with XIAP antibodies, and then incubated for a final hour at 37 degrees Celsius with an HRP-linked secondary antibody (actin was used as a housekeeping control).

### 4.9. Lung Cancer Induction in BALB/c Mice

The Animal Experimentation Ethics Committee at Chongqing University developed standards for laboratory animal care, which were followed in all animal research. An animal model of lung cancer was created by injecting B16F10 cells (1.5 × 10^6^ cells per mouse) into female BALB/c mice (18–20 g).

### 4.10. Biodistribution

GA-SNE (GA = 3 mg/kg, molar ratios of GA: XIAP siRNA = 3) was administered to mice either intravenously or by pulmonary delivery. Mice were sacrificed, and their primary organs—their hearts, lungs, livers, spleens, and kidneys—were removed at 0, 12, and 24 h after injection. An in vivo imaging system was used to investigate the fluorescence distribution following PBS washing (PerkinElmer, Waltham, MA, USA).

### 4.11. In Vivo Antitumor Therapy

Twenty female BALB/c mice (18–20 g) in total were distributed into five groups of four at random. Four doses of treatment were given on days 7, 9, 11, and 13, either by pulmonary administration or by tail-vein injection. For the pulmonary administration, the mice were administered the formulations (GA = 3 mg/kg, molar ratios of GA: XIAP siRNA = 3) in a volume of 25 L in PBS by intratracheal instillation after they had been given general anesthesia. Tail-vein injection used the same volume and concentration of the drug.

### 4.12. Histological Examination

Lung specimens were fixed with paraformaldehyde (4%), embedded in paraffin, sectioned, and stained with hematoxylin and eosin (H&E). For histopathological observations to further assess the in vivo safety of various formulations, other organs (hearts, livers, spleens, and kidneys) were collected from each group of mice, stained with H&E, and assessed by microscopy.

### 4.13. TUNEL Assay

After 30 min of 0.1% Triton X-100 incubation, tumor slices underwent two PBS washes. The TUNEL assay was carried out following the kit’s directions. Slices were photographed under a microscope after the nuclei were stained with DAPI for 10 min at room temperature.

### 4.14. Immunohistochemistry

An immunohistochemical assay was performed to evaluate the amount of Ki67. Lung sections were briefly deparaffinized by xylene and rehydrated before being submerged in the antigen retrieval solutions for 15 min. After samples were exposed to the Ki67 antibody at 4 °C for overnight incubation, they were then exposed to the secondary antibody for an additional hour. Following that, the kit’s instructions (Sangon Biotech, Shanghai, China) called for immunohistochemistry detection. Finally, a microscope was used to view the stained slices.

### 4.15. Statistical Analysis

The information was displayed as means and standard deviations. The statistical significance was examined using Student’s t-test. Statistical significance was indicated by a *p* value of 0.05 or lower.

## 5. Conclusions

Overall, we successfully developed an efficient and safe delivery system of chitosan-based cationic nanoemulsions for the co-delivery of XIAP-siRNA and GA. The system can synergistically promote tumor cell apoptosis and inhibit the malignant proliferation of lung tumors. Chitosan imparts a positive charge and biocompatibility to a co-delivery system for better siRNA protection and rapid drug release into tumor cells. XIAP siRNA can not only promote cell apoptosis but also inhibit the proliferation of tumor cells by regulating the cell cycle. In vivo studies showed that GA-SNE significantly inhibited tumor growth by pulmonary administration and elicited little toxicity in mice. According to these results, pulmonary delivery of siRNA and GA treatment using chitosan-based cationic nanoemulsions may be a potential approach to the treatment of lung tumors.

## Data Availability

Not applicable.

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
