# Peer review of "Nanoemulsion Co-Loaded with XIAP siRNA and Gambogic Acid for Inhalation Therapy of Lung Cancer"

_ijms, 2022, doi:10.3390/ijms232214294_

Round 1

Reviewer 1 Report

1. Abbreviations should be listed with their definitions (e.g. GA-CNE, GA-SNE, etc.)

2. Introduction: Scheme 1 has no description. This Scheme should be placed in the Materials and Methods section or could be a part of the graphical abstract.

3. Results:

-There is no clear explanation of the choices made at subsequent steps of the study (e.g. A549 cells treated with GA:XIAP siRNA (1:3) showed the strongest inhibition of viability. It is not clearly apparent from figure 3A.)

-Statistical data is missing in some Figures (eg. how is '****' to be interpreted?)

- Figure 4 is not referenced in the manuscript or there is a mistake in 2.5 Western blot subsection (page 5, line 187 and 191). 

4. Discussion: presented results have not been discussed, limitation is also expected to make a comment. What the next steps should be taken to achieve progress towards the pulmonary administration of GA and siRNA combined therapy in the clinical practice?

5. Materials and Methods:

-'Then, to prepare nanoemulsions (GA-274 NE), homogenize the prepared coarse emulsion using a high-pressure homogenizer.' - manufacturer and country should be indicated.

-A549 cell culture conditions, medium used, number of technical replicates in cell culture, pooling the collected cells, detailed characteristics of BALB/c mice, name of the immunohistochemical kit used to Ki67 evaluation - should be indicated in M&M section 

-How was the concentration and exposure time to Free Cy3-siRNA, GA-CNE and GA-SNE determined (page 9, line 298-299)? 

Author Response

Response to Reviewer 1 Comments

First of all, we thank both Referees for their positive and constructive comments and suggestions.

Responses to comments:

Reviewer 1

Point 1. Abbreviations should be listed with their definitions (e.g. GA-CNE, GA-SNE, etc.)

Response 1: We have added an abbreviation list as suggested (line 481-492, P17).

Point 2. Introduction: Scheme 1 has no description. This Scheme should be placed in the Materials and Methods section or could be a part of the graphical abstract.

Response 2: A graphical abstract and description of Scheme 1 have been added to the article as suggested. (line94-101, P3)

Point 3. Results:

-There is no clear explanation of the choices made at subsequent steps of the study (e.g. A549 cells treated with GA:XIAP siRNA (1:3) showed the strongest inhibition of viability. It is not clearly apparent from figure 3A.)

Response 3: We referenced an article on the use of antibody drugs in combination with the chemotherapy drug PTX (https://doi.org/10.1021/acs.nanolett.1c01210),the authors optimized the ratio of PTX and aCD47 by culturing PTX liposomes with different ratios of PTXaCD47 (1:3, 1:2, 1:1, 2:1 and 3:1) with MDA-MB-231 cells and calculating the survival rate of MDA-MB-231 cells treated with different ratios. We optimized the ratio of our GA and XIAP siRNAs by referring to his approach, and our previous experimental results showed that the anti-tumor effect was slightly better when GA: XIAP siRNA=4 than GA: XIAP siRNA=3. However, considering that Gambogic acid is a small molecule with certain toxicity, we finally chose GA: XIAP siRNA=3 for the sake of safety and efficacy. (line179, P7)

-Statistical data is missing in some Figures (eg. how is '****' to be interpreted?)

Response 3: Previously we used One-Way ANOVA for each concentration (0.1-0.6 μg) of CS-SNE, GA-CNE and GA-SNE. In addition, we used Two-Way ANOVA to recalculate the data and mark them in the figure. (line179, P7)

- Figure 4 is not referenced in the manuscript or there is a mistake in 2.5 Western blot subsection (page 5, line 187 and 191). 

Response 3: This was a clerical error and we have corrected it (line 206, line 210, P8).

Point 4. Discussion: presented results have not been discussed, limitation is also expected to make a comment. What the next steps should be taken to achieve progress towards the pulmonary administration of GA and siRNA combined therapy in the clinical practice?

Response 4: The discussion section has been added as suggested. (line429-448, P16)

Point 5Materials and Methods:

-'Then, to prepare nanoemulsions (GA-274 NE), homogenize the prepared coarse emulsion using a high-pressure homogenizer.' - manufacturer and country should be indicated.

Response 5: Manufacturers and countries have been added as recommended. (line307, P13)

-A549 cell culture conditions, medium used, number of technical replicates in cell culture, pooling the collected cells, detailed characteristics of BALB/c mice, name of the immunohistochemical kit used to Ki67 evaluation - should be indicated in M&M section 

Response 5: Cell, animal information and kit names have been added as suggested. (line288-289, line 292-299, P13)

-How was the concentration and exposure time to Free Cy3-siRNA, GA-CNE and GA-SNE determined (page 9, line 298-299)? 

Response 5: The concentration of Coumarin-6 we refer to the work of Kebebe et al ( DOI: 10.2147/IJN.S202424). In addition, we determined the concentration and incubation time of siRNA by referring to the siRNA product specification and the work of Seidel et al ( DOI: 10.1021/acsabm.0c00503).

Reviewer 2 Report

In this paper, the author described a pulmonary delivery system using nanoemulsion, the idea is of importance and the results supported the conclusions. However, some points may need to be clarified. My detailed comments follow.

1. In the intracellular uptake study, the authors used Ce-6 as the fluorescent probe, however, the characterization of GA-SNE/CNE with Ce-6 should be proven to be the same as GA-SNE/CNE.

2. Line 174-175, "compared to siRNA alone (78.12%)", however, it is not consistent with the results in Figure 3C which siRNA alone group showed apoptosis less than 10%.

3. No label in Fig. 3B, and the label in Fig. 3C is not clear.

4. Fig. 4, the band of protein is hardly seen, I suggest the authors to repeat the western blot

5. Fig. 5B, there is no indication of each tissue.

Author Response

Response to Reviewer 2 Comments

First of all, we thank both Referees for their positive and constructive comments and suggestions.

Responses to comments:

Reviewer 2

Point 1. In the intracellular uptake study, the authors used Ce-6 as the fluorescent probe, however, the characterization of GA-SNE/CNE with Ce-6 should be proven to be the same as GA-SNE/CNE.

Response 1: Sorry, due to a clerical error we abbreviated Coumarin 6 to Ce6, which has been corrected in the text. In addition, relevant data on C6-CNE characterization have been uploaded as suggested. (line134, P4, line139, P5)

Point 2. Line 174-175, "compared to siRNA alone (78.12%)", however, it is not consistent with the results in Figure 3C which siRNA alone group showed apoptosis less than 10%.

Response 2: Thank you for your suggestion, the text in the article was revised for unclear description. (line193, P7)

Point 3. No label in Fig. 3B, and the label in Fig. 3C is not clear.

Response 3: The markings of Fig. 3B and Fig. 3C have been modified as suggested. (line180, P7)

Point 4. Fig. 4, the band of protein is hardly seen, I suggest the authors to repeat the western blot

Response 4: The western blot has been repeated as recommended. (line212, P8)

Point 5. Fig. 5B, there is no indication of each tissue.

Response 5: Each tissue has been labeled as recommended. (line228, P11)

Round 2

Reviewer 2 Report

In the revised MS, the authors corrected and clarified the points raised by reviewers, I think it is now acceptable, however, the English need to be polished and carefully checked.